# Increased Antimicrobial Resistance of MRSA Strains Isolated from Pigs in Spain between 2009 and 2018

**DOI:** 10.3390/vetsci6020038

**Published:** 2019-04-04

**Authors:** Rossana Abreu, Cristobalina Rodríguez-Álvarez, María Lecuona, Beatriz Castro, Juan Carlos González, Armando Aguirre-Jaime, Ángeles Arias

**Affiliations:** 1Department of Preventive Medicine and Public Health. Universidad de La Laguna. Campus de Ofra S/N., 38071 Santa Cruz de Tenerife, Spain; rabreu@ull.edu.es (R.A.); crrodrig@ull.edu.es (C.R.-Á.); 2Microbiology and Infection Control Service of the University Hospital of the Canary Islands (HUC), 38004 Tenerife, Spain; mlecuona2005@yahoo.es (M.L.); bcastrohdez@yahoo.es (B.C.); 3Canary Islands Health Service, Santa Cruz de Tenerife, 38004 Canary Islands, Spain; ecoladera@gmail.com; 4Institute of Care Research, Nurses’ Association, 38001 Santa Cruz de Tenerife, Spain; armagujai@gmail.com

**Keywords:** MRSA, SCC*mec*, ST398, swine, antibiotic resistant

## Abstract

The problem of emerging resistant microorganisms such as Methicillin-Resistant *Staphylococcus aureus* (MRSA) associated to livestock is closely linked to improper use of antimicrobial agents. The aim of this study is to find out the prevalence and characteristics of these strains, as well as their evolution in healthy pigs on the Island of Tenerife, Spain. Between October 2009 and December 2010, 300 pigs from 15 wean-to-finishing farms were screened. Between 1 September 2017 and 31 March 2018, a new sampling was performed collecting 125 nasal swabs from pigs belonging to the same farms and under the same conditions as the previous study. MRSA antibiotic resistant patterns were studied. Results: Prevalence of MRSA isolates was 89.6%. All isolates belonged to Sequence Type 398 (ST398), a livestock related strain. All strains studied were resistant to beta-lactamic non-carbapenemic antibiotics and sensitive to teicoplanin, linezolid, vancomycin, rifampicin, quinupristin-dalfospristin, and mupirocine. Between 2009/2010–2017/2018 a significant increase in resistance to gentamicin, tobramycin, trimethoprim-sulfomethoxazole, clindamycin, Fosfomycin, and tigecycline antibiotics was observed in isolated MRSA strains compared to the previous period. We consider a major control and surveillance program of antibiotic use in veterinary care is needed in order to reduce the presence of MRSA strains in livestock and control this significant multi-resistance increase.

## 1. Introduction

Concerns about the acquisition of antibiotic resistant genes in human and animal pathogens is one of the main public health issues. Thus, the World Health Organisation [1] regards as a priority objective the appropriate use of antimicrobials in human and animal clinical practice. 

The problem of the emergence of resistant microorganisms such as Methicillin-resistant *Staphylococcus aureus* (MRSA) associated to livestock is closely linked to high use of antimicrobial agents in veterinary care, and international trade of food of animal origin which can encourage the spread of resistant strains. Drug-resistant bacteria may circulate in human and animal populations trough food, water and the environment, and its transmission is encouraged by trade, travel, human migration and transhumance [2]. Most animals included in the food chain may be colonized with *S. aureus.* The importance of livestock associated MRSA (LA-MRSA) strains has been recognized since 2005 [3]. MRSA strains have been detected in food production animals, such as swine, cattle, chicken, and other animals [4,5,6,7,8,9], as well as in various types of food products including raw chicken meat, retail pork and beef, milk and dairy products [10,11,12,13,14], and fishery products [15], in farm environments, where transmission can even occur between different animals [16,17]. Farmers easily acquire these MRSA strains, which is a work-related health problem and they can also spread them to the general population [6,9,18,19,20,21]. While spread to the general population is possible, the LA-MRSA is only seen in limited percentages in the community and especially in hospital settings. In the European Union, swine is the largest livestock population and in 2016, Spain held the largest livestock population in the EU-28. The largest swine populations were recorded in Spain and Germany (30 and 27.6 million head respectively) [22].

We believe it is important to identify the presence of MRSA in this type of livestock in our country. The aim of our study is to find out and assess the prevalence and characteristics of these strains, as well as their evolution in healthy pigs on the Island of Tenerife, Spain.

## 2. Materials and Methods

### 2.1. Collection of Samples

A cross-sectional prevalence study was conducted. Between October 2009 and December 2010, 300 pigs from 15 wean-to-finishing farms were screened [6]. Later, between 1 September 2017 and 31 March 2018, a new sampling was performed, whereby 125 nasal swabs were collected from pigs belonging to the same farms and under the same conditions as the previous study. The sampling was done by an expert veterinary of the Public General Direction, just before the animals were slaughtered and while they were unconscious (carbon dioxide chamber and subsequent bleeding). The study has been authorized by the Health Public General Direction of Health Service of Canary Islands Government and The Canary Islands Research and Health Foundation (FUNCIS-ENF08/09).

### 2.2. Isolation and Identification of Bacteria

Nasal swabs were incubated in brain heart infusion broth (BHI) with 7% NaCl for 18–24 h at 37 °C. Then, 10 µL of this broth was plated onto selective chromID^®^MRSA agar culture (bioMérieux^®^, Marcy l’Etoile, France), and incubated for 48 h at 37 °C. MRSA colonies were preliminarily identified as characteristic green malachite colored round colonies and purified on Columbia agar plates with 5% sheep blood (bioMérieux^®^). Isolates were confirmed as *S. aureus* by Gram stain appearance catalase test and coagulase test agglutination (Slidex^®^ Staph Plus; bioMérieux^®^). Species’ identifications were confirmed by Vitek^®^2 Automated Microbiology System with a Gram positive identification card (bioMérieux^®^). Methicillin resistance was confirmed by testing the presence of penicillin-binding protein A (PBP2a) (MRSA-screen; Denka Seiken Co™, Tokyo, Japan) and detecting the presence of the *mecA* gene by Real Time PCR (IQ™5; Bio-Rad, Hercules, CA, USA). The mecA gene (496bp) was amplified with the following primers: *mecA*- 1: 5′-GGG GTG GTT ACA ACG TTA CAA G-3′ *mecA*- 2: 5′-AGT TCT GCA GTA CCG GAT TTG C-3′. *S. aureus* ATCC 29213 was used as the reference strain. The primers were marked with IQ SYBR Green Supermix (BioRad^®^, Madrid, Spain).

### 2.3. Molecular Typing of MRSA

*DNA macro-restriction analysis and pulsed-field gel electrophoresis.* Isolation of chromosomal DNA was performed as described by Smith et al. [23]. The enzyme used for the macro-restriction was *ApaI*.

block II: 6 v/cm, 12 h; 15–40 s. The results were interpreted according to the criteria described by Tenover et al. [24]. *Multilocus Sequence Typing* (MLST). All MRSA isolates were analyzed by MLST, as described by Enright et al. [25]. To extract bacterial DNA, a DNeasy Blood & Tissue Kit (Qiagen, Sarasota, FL, USA) purification kit was used. The Allelic profiles and sequence types were assigned according to the *S. aureus* MLST database (http://www.mlst.net). *Staphylococcal Chromosome Cassette (SCC) mec typing*. The different pulsotypes and the subtypes obtained were analyzed by SCC*mec* typing, as described by Milheirico et al. [26].

### 2.4. Antimicrobial Susceptibility Testing

Antimicrobial susceptibility was determined by the broth microdilution method (Vitek^®^ 2 system, bioMérieux^®^) by using AST P626. *S. aureus* ATCC 29213 was used as a reference strain. Strains were tested for susceptibility to aminoglucosides: Gentamicin (GM), tobramycin (TM); beta-lactamic: Bencylpenicillin (PG), oxacillin (OXA); glucopeptides: Teicoplanin (TEI), vancomycin (VA); quinolone: Levofloxacin (LVX); lincosamides: Clindamycin (CC); Macrolides: Erythromycin (E); rifamycin: Rifampicin (RI); bacteriostatics trimethoprim-sulfametoxazole (SXT), fusidic acid (FA); streptogramins: Quinupristin–dalfopristin (QDA); oxazolidinone: Linezolid (LZ), glycylcycline: Tigecicline (TGC); phosphate: Fosfomycin (FM); nitrofurane: Nitrofurantoin (NI); monoxycarbolic acid: Mupirocine (MUP). The breakpoints used were those established by the Clinical and Laboratory Standards Institute (CLSI) Guidelines [27].

### 2.5. Statistical Analysis

The characteristics of the sample are described with the relative frequency of the categories that make up nominal variables. Frequencies’ comparisons were performed with Fisher’s Exact Test at a significance level *p* ≤0.05. Calculations were performed with the IBM Co™ statistical processing package for PC in operating environment Windows SPSS 21.0 (SPSS, Chicago, Illinois, USA)

## 3. Results

Prevalence of the MRSA isolates was 89.6% (112/125). Multilocus sequence typing (MLST) confirmed all isolates belonged to Sequence Type 398 (ST398). All strains studied were resistant to beta-lactamic non-carbapenem antibiotics and sensitive to teicoplanin, linezolid, vancomycin, rifampicin, quinupristin-dalfospristin, and mupirocine.

Table 1 shows the percentage of resistance to antibiotics tested in the two different time periods studied. The table shows a considerable resistance increase to gentamicin, tobramycin, trimethoprim-sulfomethoxazole, clindamycin, Fosfomycin, and tigecicline antibiotics during 2017‒2018, compared to the previous period. Only in the case of erythromycin was a significant reduction in the percentage of resistant strains observed.

Table 2 shows MRSA resistance patterns in both periods. A total of 29 resistance patterns were obtained, presenting differences between them with 20 different patterns for those isolated between 2009‒2010 and 13 for the strains isolated between 2017‒2018. In this last period, PG + OXA + GM + TM + CC + STX resistance pattern was obtained in almost half of the strains studied. However, in the previous period, the most common patterns were single resistance to beta-lactamic PG + OXA, identified in almost a fifth of the strains studied and PG + OXA + E + CC pattern, found in 17.2% of the strains studied.

*SCCmec* typing of our MRSA pig isolates revealed the presence of type IV and type V in both periods, type V being the most predominant in the first period of the study (62.5% of isolates) and type IV in 2017‒2018 period (61.6%), this being a significant difference (*p* < 0.001).

## 4. Discussion

In our study, we found a high prevalence of MRSA strains resistant to various groups of antibiotics. The prevalence of LA‒MRSA of the ST398 lineage in pig was 89.6%, which is similar to the result obtained in the previous study during the 2010‒2011 period (85.7%) [6]. Another Spanish study performed in pigs and piglets in a slaughterhouse demonstrated high MRSA colonization prevalence, finding ST398 and ST97 strains [28]. Reynaga et al. [9] in Catalonia found a prevalence of 46.0%, though the sampling was performed in farms. Various studies at a worldwide level have found variable MRSA prevalence percentages in this kind of livestock. The LA-MRSA CC398 strain has been described in many countries, in Europe and worldwide, with different prevalence in pork livestock and workers of the sector [11,21,29,30,31,32].

In our study, it should be emphasized that MRSA strains show resistance to other antimicrobial groups such as aminoglycosides, tetracyclines, lincosamides, macrolides, etc. This has been described by several authors [9,26,33,34]. Analysis of the susceptibility status of MRSA pig isolates revealed that in 2009, 50 isolates (19.5%) were susceptible to all antibiotics tested, except to betalactamic. However, in the last period studied, resistance patterns included a higher number of antibiotics.

Spain was the highest consumer country of veterinary antimicrobials in 2015 within the countries that submitted data to the European Surveillance of Veterinary Antimicrobial Consumption (ESVAC) (The European Medicines Agency) [35]. Antibiotics are often used massively without diagnostic tests, metaphylaxis being widespread with no appropriate justification. Since 2014, our country has developed a Strategy and Action Plan to reduce the risk of selection and dissemination of antibiotic resistance, emphasizing the importance of surveillance and control of rational use of antibiotics in animal healthcare [36]. However, our study shows a significant resistance increase instead of the reduction expected from the implementation of the said plan, although still under way. Accordingly, Dierikx et al. [37] indicated a prevalence increase of MRSA nasal carriers in pig screened in Dutch slaughterhouses, despite the reduction in the use of antimicrobials at a national level, arguing this antimicrobial control had not yet had an effect on MRSA carrier rates in pigs.

In our study, the chromosomal cassette has changed, finding significant differences as there have been a higher number of *SCCmec* type IV carrier strains in the last period than in the previous one. We also observed that the resistance pattern including more antibiotics corresponded to those MRSA isolates having *SCCmec* type IV. Gómez-Sanz et al. [28], also indicated that the genotype of multi-resistance is generally associated with the presence of SCCmec type IV.

One limitation of our study was that the swine included always came from intensive farms and we did not have access to information on their antimicrobial use.

In conclusion, we consider a major control and surveillance program of antibiotic use in veterinary care is needed to reduce the presence of MRSA strains in livestock, as well as to control this significant multi-resistance increase with greater vigilance on the part of health authorities to comply with the strategies and plan of action established.

## Figures and Tables

**Table 1 vetsci-06-00038-t001:** Statistical significance of antimicrobial resistance according to period studied.

Antimicrobial	Number and Percentages of Resistant Isolates		Significance (*p*)
	Period Studied		
	2009–2010	2017–2018	
Gentamicin	98 (38.3)	69 (61.6)	<0.001
Tobramycin	101 (39.5)	74 (66.1)	<0.001
Trimethoprim-sulfomethoxazole	99 (38.7)	69 (61.6)	<0.001
Levofloxacin	34 (13.3)	11 (9.8)	Non-significant
Erythromycin	86 (33.6)	18 (16.1)	<0.001
Clindamycin	129 (50.4)	86 (76.8)	<0.001
Fosfomycin	1 (0.4)	33 (29.5)	<0.001
Tigecycline	0 (0)	14 (12.5)	<0.001

**Table 2 vetsci-06-00038-t002:** Resistance patterns.

Resistance Pattern	2011		2017	
	Number	%	Number	%
PG + OXA	50	19.5	0	0
PG + OXA + E + CC	44	17.2	0	0
PG + OXA + GM + TM + SXT	41	16.0	6	5.3
PG + OXA + GM + TM + E + CC + SXT	34	13.3	0	0
PG + OXA + LVX + CC	19	7.4	0	0
PG + OXA + LVX	13	5.1	0	0
PG + OXA + CC	13	5.1	0	0
PG + OXA + SXT	11	4.3	0	0
PG + OXA + GM + TM	8	3.1	1	0.9
PG + OXA + GM + TM + CC + SXT	8	3.1	54	48.2
PG + OXA + E + CC + SXT	3	1.2	0	0
PG + OXA + GM + TM + CC	3	1.2	7	6.2
PG + OXA + TM + LVX + SXT	2	0.8	0	0
PG + OXA + GM	1	0.4	0	0
PG + OXA + TM	1	0.4	0	0
PG + OXA + GM + TM + E + CC + FM + SXT	1	0.4	0	0
PG + OXA + LVX + E + CC	1	0.4	0	0
PG + OXA + GM + TM + E + CC + NI	1	0.4	0	0
PG + OXA + TM + E + CC	1	0.4	0	0
PG + OXA + GM + TM + E + CC	1	0.4	0	0
PG + OXA + CC + FM + NI	-	-	20	17.8
PG + OXA + E + LVX + TGC	-	-	6	5.3
PG + OXA + E + TGC + FM	-	-	5	4.5
PG + OXA + E + TGC + FM + SXT	-	-	3	2.7
PG + OXA + E + CC + FM + SXT	-	-	3	2.7
PG + OXA + GM + LVX + STX	-	-	3	2.7
PG + OXA + TM + LVX	-	-	2	1.8
PG + OXA + E + CC + FM	-	-	1	0.9
PG + OXA + GM + TM + CC + FM	-	-	1	0.9

PG: Bencylpenicillin; OXA: Oxacilin; GM: Gentamicin; TM: Tobramicin; E: Erithromycin; CC: Clindamycin; FM: Fosfomycin; NI: Nitrofurantoin; LVX: Levofloxacin; TGC: Tigecycline; SXT: Trimethoprim/Sulphamethoxazole.

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
