# Peer review of "Increased Antimicrobial Resistance of MRSA Strains Isolated from Pigs in Spain between 2009 and 2018"

_vetsci, 2019, doi:10.3390/vetsci6020038_

Round 1

Reviewer 1 Report

This is an interesting paper. In my opinion, you should include some description of methods used in the 2010-11 study

Results in table 2 may be confusing. You could try to group resistance pattern into antibiotic family to clarify and made easier comparisons among the two periods and avoid comparison if the antibiotic tested are not the same.

Author Response

We have followed the instructions of the editor

Reviewer 2 Report

Interesting study on MRSA in pigs. Suggest some minor improvements: 

Abstract:

-  Line 19-21: suggest to add a bit more detail about the 2012 publication namely that in 2009-2010 also nasal swabs were used and that prevalence then was 85,7% . 

- Line 25, would be easier if you write  'Between 2009/2010-2017/2018, a significant ...'. 

 - Line 28-29: suggest a surveillance program both on antibiotic use but also on amr. 

 Keywords

- green block to be removed. 

Introduction

Line 35, suggest to also include a newer reference namely the 2017 WHO guidelines on use of medically important antimicrobials in food-producing animals

Line 47: 'they can spread them to the general population'. Maybe worth specifying that while spread to the general population is possible, the LA-MRSA is only seen in limited percentages in community and especially in hospital settings. 

Line 49: think poultry is most numerous livestock 

Line 50: please update to the 2017 data, they are available on Eurostat. 

Discussion 

Line 130 'which is similar to the results obtained in the previous study - suggest to repeat the main results, so that people can easily compare. One difference was that the original study also looked at MRSA in humans, which is not the case in this one. 

Line 133 A new report just released was the EFSA/ACEC report on AMR in zoonotic and dincicator bacteria from humans, animals and food 2007. It also has a whole chapter on MRSA . I recognise this report came out after you put this article for peer review, but seen the importance of this report it would be good when reviewing your article, to use the opportunity to build in this very important report. 

Line 136 - suggest also to take into account 

Christiane Cuny, Robin Köck, Wolfgang Witte Livestock associated MRSA (LA-MRSA) and its relevance for humans in Germany, International Journal of Medical Microbiology, Volume 303, Issues 6–7, 2013,

Line 142 ESVAC data - please use the latest data which where published in October 2018. 

Line 144-145: please add reference 

Line 150-152: in contradiction to the Dutch results, the Norwegians had success in controlling MRSA. Maybe good to also add something on this. 

Line 153-157 - again here it would be interesting to also build in some key results of the EFSA/ECDC AMR report (EFSA Journal 2019; 17(2):5598

Line 158-159: replace 'although' by 'and that'  One limitation of our study was that ... farms and that we did not have information ...

Line 161 - suggest a surveillance program both on antibiotic use but also on amr. 

Author Response

First of all, we would like to thank for your helpful comments that have enhanced our article.

Summary

Line 19-21- Due to the limitation of words in the summary it has been added in the article.

Line 25: the text has been modified

Line 28-29: Due to the limitation of words in the summary it has been added in the conclusion of the article

Introduction

Line 35: We changed the references

Line 47: It has been included

Line 49: It is correct according to the consulted data.

Line 50: The data has been updated

Discussion

Line 130: It has been modified

Line 133: It has been included

Line 136: It has been included

Line 142: It has been included

Line 144-145: The appointment has been modified

Line 153-57: We have not found a reference in this report

Line 158-159: It has been replaced

Line 161: It has been included

Reviewer 3 Report

Major comments

Results –In general the results sections needs improvement.

Line 102 – The frequency of MRSA positive isolates should be outlined along with the percentages.

Discussion – The bigger picture for the need of the study can be improved.

Were there any methicillin sensitive Staphylococcus aureus among the tested animas or samples? If so what percentage? How were the frequencies and percentages between the two studies?

Line 71- 73 – There are two PCRs here and one real time and one seem to the conventional PCR.  The sentence implies that a real time PCR was performed with following primers. Consider outlining the primer and probe sequences for the real time PCR and primers and conformation of PCR products (gel electrophoresis) for the conventional PCR.

Minor comments

Line 38 – Inadequate use has different meanings at different context, authors need to correct this accordingly.

Line 43- LA MRSA? – Nowhere in the manuscript this is abbreviated.

Line 62- BHI agar or broth?

Author Response

First of all, we would like to thank for your helpful comments that have enhanced our article.

Line 102: It has been included.

Discussion:

We have detected MRSA directly, with a specific medium agar.

Line 71-73: We have not used conventional PCR. We have added “the primers were marked with ...... “  in material and methods.

Line 38: It has been modified

Line 43: it has been modified

Line 62: The word “Broth” has been included